# NOT ALL LLM REASONERS ARE CREATED EQUAL

## ABSTRACT

We study the depth of grade-school math (GSM) problem-solving capabilities of LLMs. To this end, we evaluate their performance on pairs of existing math word problems together so that the answer to the second problem depends on correctly answering the first problem. Our findings reveal a significant reasoning gap in most LLMs, that is performance difference between solving the compositional pairs and solving each question independently. This gap is more pronounced in smaller, more cost-efficient, and math-specialized models. Moreover, instruction-tuning recipes and code generation have varying effects across LLM sizes, while finetuning on GSM can lead to task overfitting. Our analysis indicates that large reasoning gaps are not because of test-set leakage, but due to distraction from additional context and poor second-hop reasoning. Overall, LLMs exhibit systematic differences in their reasoning abilities, despite what their performance on standard benchmarks indicates.

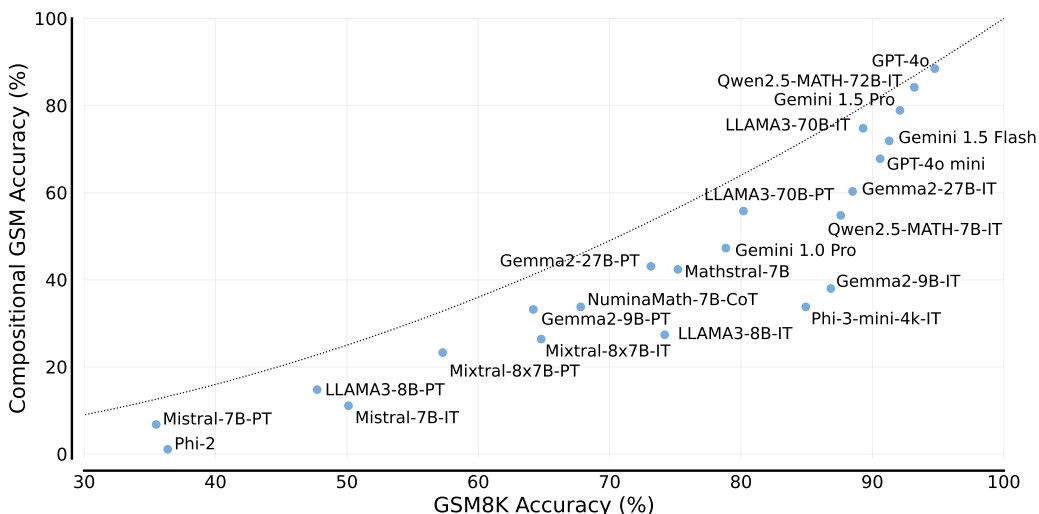

Figure 1: **Reasoning Gap:** Most models demonstrate a noticeable gap between their reasoning performance on GSM8K and compositional GSM, in which pairs of GSM8K test questions are chained together so that the answer of the first question ($Q_1$) is a variable in the second one ($Q_2$). The model is required to correctly answer both questions to solve the problem. If a model has an accuracy of $S_1$ on the $Q_1$ set, and $S_2$ on $Q_2$ set, then the expected compositional GSM accuracy is $S_1 \times S_2$. The x-axis corresponds to the geometric mean $\sqrt{S_1 \times S_2}$, labeled GSM8K accuracy for simplicity. The trend-line $y = x^2$ is the expected compositional GSM accuracy.

## 1 INTRODUCTION

The strong performance of large language models (LLMs) on high-school and college-level math reasoning benchmarks (OpenAI, 2023b; Google, 2024; AI@Meta, 2024), has led to the common belief that LLMs have "mastered" grade-school math, particularly as measured by the GSM8K benchmark (Cobbe et al., 2021). This apparent mastery of grade-school math problems raises a deeper question: do LLMs truly grasp the underlying concepts or do they mostly rely on superficial pattern recognition? For example, a recent examination on private "held-out" grade-school

---

Let X be the answer to the $\mathbf{Q_1}$:

$\mathbf{Q_1}$: There are 27 unicorns left in the world. One third of them are in the Scottish Highlands. Two thirds of the Scottish unicorns are female. How many female Scottish unicorns are there?

Solve it and use the value of X to solve $\mathbf{Q_2}$. Explain your answer step by step.

$\mathbf{Q_2}$: Zack's locker is half as big as Timothy's locker. Peter's locker is 1/4 as big as Zack's locker. If Peter's locker is X cubic inches, how big is Timothy's locker in cubic inches?

---

Figure 2: **Example Problem from the Compositional GSM test**. The answer of Question-1 ($\mathbf{Q_1}$) is a variable X in Question-2 ($\mathbf{Q_2}$). The model has to be able to solve the first question correctly in order to solve the second question. The new final answer of $\mathbf{Q_2}$ is calculated by modifying its code-form solution and executing it. We used a modified version of the code-form solutions from Gao et al. (2023). Question-1 and the number to modify in Question-2 are chosen to have a new final answer which is a positive integer not too far from the old answer of Question-2.

problems (Zhang et al., 2024) reveals that while state-of-the-art LLMs show minimal signs of over-fitting, some open-weights models show systematic overfitting, possibly due to test-set leakage.

In this work, we perform a case study to probe the brittleness of their reasoning abilities and to evaluate how well LLMs can combine learned concepts to solve new problems (Hupkes et al., 2020) To do so, we introduce *Compositional GSM*, a two-hop version of GSM8K at the same math difficulty level, where each problem chains two test questions together such that the answer to the first question is used as a variable in the second question (Figure 2). As LLMs can easily solve grade-school math problems, they should also be capable of solving combinations of those problems. As such, we measure the gap between their performance on solving the questions individually and on compositional GSM. Specifically, we benchmark frontier open-weights and closed LLMs, including Gemini, Gemma2, LLAMA3, GPT, Phi, Qwen2.5, and Mistral families. Here are our key findings:

- Most models exhibit a clear gap between their performance on GSM8K and compositional GSM (Figure 1,3), which undermines their reliability and reasoning ability.
- This reasoning gap is particularly evident in small, more cost-efficient (Figure 4), and math-specialized models (Figure 6), reducing their utility in practice.
- Despite similar settings, instruction-following tuning impacts LLMs of varying sizes in significantly different ways (Figure 5), calling for re-examination of standard training recipes.
- Finetuning with either human or synthetic data on GSM8K problems results in task-specific over-fitting with longer training (Figure 7).
- Smaller models benefit more from generating code solutions rather than natural language to solve compositional problems, emphasizing systematic differences in reasoning abilities (Figure 8).
- Our analysis (in §3.6) indicates that large reasoning gaps are not due to test-set leakage, but the result of distraction from additional context and poor second-hop reasoning

Our objective is not simply to introduce yet another reasoning benchmark, but to provide a case study for deeper insights into LLM' reasoning and a reassessment of how we evaluate these abilities.

## 2 COMPOSITIONAL GRADE-SCHOOL MATH (GSM)

Each question in compositional GSM consists of two questions, Question-1 and Question-2, from a subset of 1200 examples of the original GSM8K test set. The answer of Question-1 is a variable in Question-2, which is referred as $X$, as shown in Figure 2. The answer of Question-2 is obtained by substituting $X$ and solving it. The choice of Question-1 and the number to modify and replace with $X$ in Question-2 was made in a way such that the new final answer of Question-2 is different from its old final answer, and is a positive integer not too far from the old final answer. To obtain the new final answer of Question-2 automatically, we replace a number in the code-form solution of Question-2. Our design choices ensured that the test set of compositional GSM and original GSM8K have similar final answer (magnitude) distributions (see Figure A.1). To make sure that the modified questions are sensible and logical, we generated 16 candidate solutions per modified question from GPT-4o and Gemini 1.5 Pro. We filtered those questions for which less than 4 (out of 16) agree with

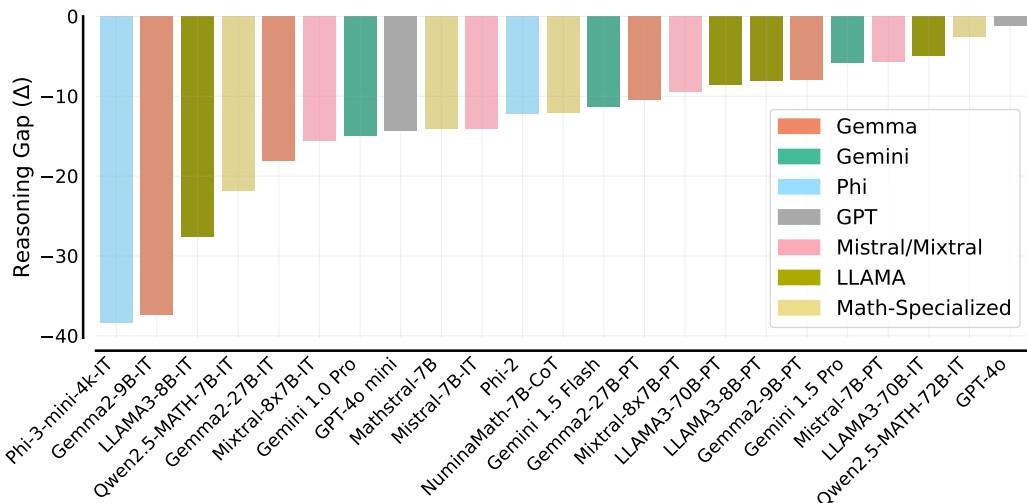

Figure 3: **Reasoning Gap** of notable open-weights and closed-source LLMs. Smaller, more cost-efficient and math specialized models have a bigger gap. See Figure 1 for GSM and compositional GSM accuracy.

the expected final answer from code execution. We checked these questions manually and modified them if needed so that they are logical (about 25% of questions).

**Reasoning Gap**. Question-1 and Question-2 in our compositional queries are from the original GSM8K test split, and the modified test split respectively. Assuming that a model has an accuracy of $S_1$ and $S_2$ on these splits, it is expected for it to have an accuracy of $S_1 \times S_2$ on the compositional split $\mathcal{D}_{\text{comp}}$. We report the following as the compositional reasoning gap score,

$$\text{Reasoning gap} : \Delta = S_{\text{comp}} - S_1 \times S_2 \qquad (1)$$

where $S_{\text{comp}}$ is the test accuracy of the model on $\mathcal{D}_{\text{comp}}$.

## 3 EXPERIMENTS & RESULTS

**Setup** We evaluate each model on three test sets: **1) the original GSM8K test split**, **2) the modified GSM8K test split** which are the questions with X being substituted, and **3) the compositional GSM** test set. Each test set has 1200 examples. Following Zhang et al. (2024), we evaluate all LLMs with a standard 8-shot prompt (Appendix D) for the original and modified GSM8K test splits. We also created a similar 8-shot prompt (Appendix E) for the compositional GSM questions. No elaborate prompting method is needed with this format. We evaluate GPT-4o, GPT-4o mini (OpenAI, 2023a), LLAMA3-70B and 8B (PT and IT) (AI@Meta, 2024), Phi 2, Phi-3-mini-instruct (Abdin et al., 2024), Gemini 1.0, 1.5 Flash and 1.5 Pro (Google, 2023; 2024), Gemma2 9B and 27B (PT and IT) (Gemma Team et al., 2024), Mistral-7B (PT and IT), Mixtral-8x7B (PT and IT) (Jiang et al., 2024), and math-specialized LLMs including Numina-7B (Beeching et al., 2024), Mathstral-7B, Qwen2.5-7B and Qwen2.5-72B (Yang et al., 2024a). All models are sampled with temperature 0, and pass@1 (Chen et al., 2021) is used to measure the performance on each test split. Some of the models required a preamble prefixed to the 8-shot prompt for desired output formatting (Appendix B). We test both cases and report the best performance for each model.

We find that most LLMs fall below expectation on compositional GSM, exhibiting a large reasoning gap as shown in Figure 3. Specifically, cost-efficient and smaller LLMs exhibit a much larger gap than closed-source frontier LLMs, which we examine in details in the following sections.

### 3.1 COST-EFFICIENT AND SMALLER LLMS REASON DIFFERENTLY

The reasoning abilities of cost-efficient LMs has been rapidly improving over time, as evaluated using standard benchmarks (Bansal et al., 2024). For example, GPT-4o mini and Gemini 1.5 Flash both achieve above 90% accuracy on GSM, while priced $25-35\times$ cheaper than GPT-4o and Gemini

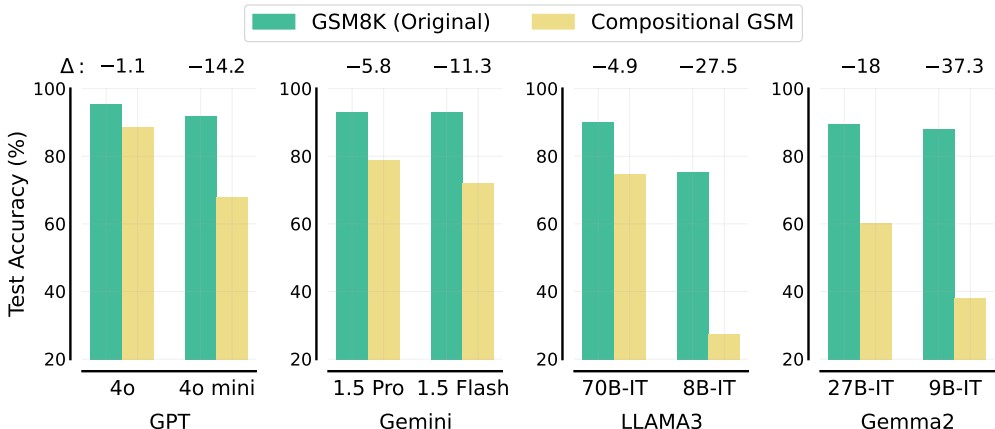

Figure 4: **Cost efficient LLMs reason differently:** showing four family of models, each having a high-cost and low-cost option. The numbers above the bars represents the reasoning gap defined in Eq 1. Although the cheaper models perform similarly on the original GSM8K test, they show a significant decline in performance on the compositional GSM test.

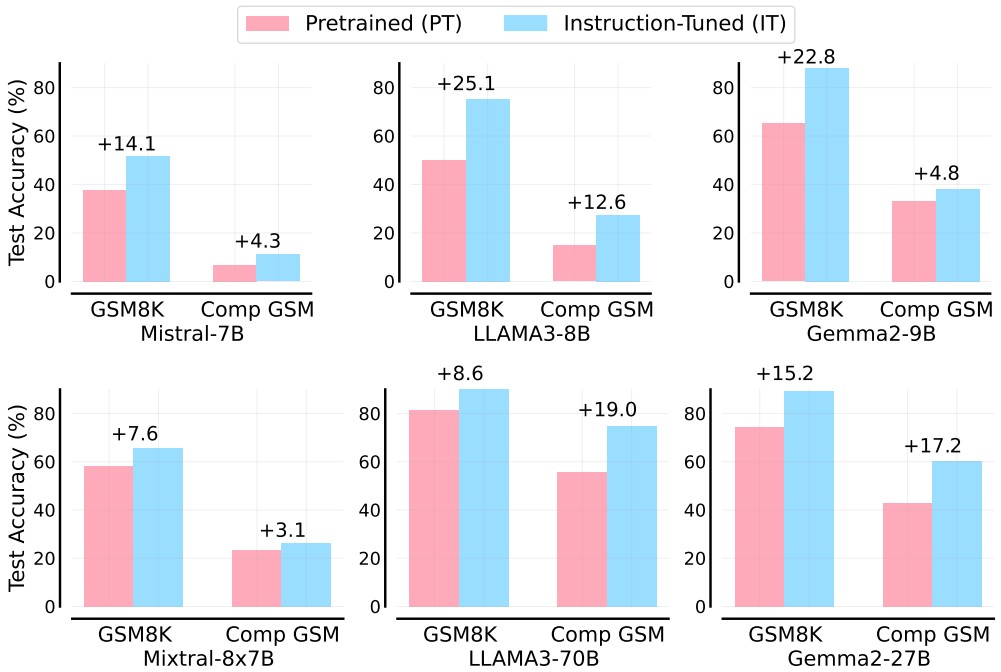

Figure 5: **Impact of Instruction-Tuning on Compositional GSM**. We compare pretrained and instruction-following tuned variant of models from Mistral, LLAMA3 and Gemma2 families. Numbers above bars represent improvements from instruction-tuning on each set. For smaller models **(top)**, we observe that instruction-tuning results in substantial improvements on the original GSM8K test set, but a much smaller improvement on the compositional GSM test. However, this pattern does not typically hold for larger models **(bottom)**.

1.5 Pro respectively. This progress could be attributed to several factors, such as better data mixtures (AI@Meta, 2024), and knowledge distillation (Team et al., 2024; Agarwal et al., 2024). To this end, we investigate whether these reasoning gains on GSM8K still persist on compositional GSM.

We study four family of models, each comprising both a high-cost and low-cost option, where cost is measured via parameter count or API pricing. Figure 4 shows the original GSM8K test split performance and compositional GSM performance for all models. While cheaper models perform comparably or slightly worse on the original GSM8K test, they exhibit a $2 - 12\times$ worse reasoning gap on compositional GSM. This gap is particularly striking for GPT-4o mini, which nearly matches

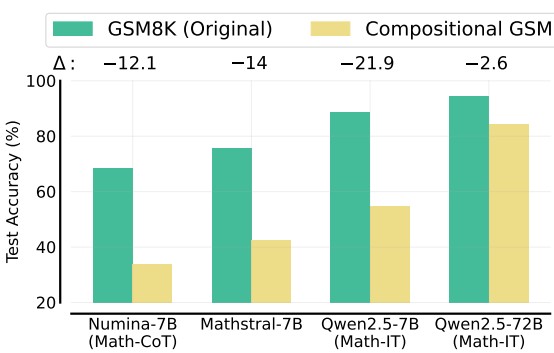
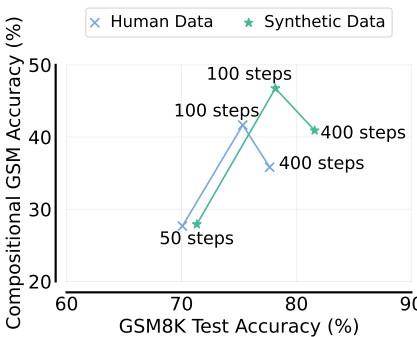

Figure 6: **Math-Specialized LLMs on Compositional GSM**. We evaluate the performance of three models specifically designed for math problem-solving to explore whether extensive specialized training in mathematics can bridge the reasoning gap observed among models of similar size or family. Surprisingly, we find that such math-specialized LLMs, particularly the smaller models, exhibit similar reasoning gaps and signs of overfitting to standard benchmarks.

Figure 7: **Overfitting with supervised fine-tuning**. We finetune Gemma2 27B on the original GSM8K training solutions, and self-generated solutions. In both settings, after 100 training steps, compositional GSM test performance drops while GSM8K test performance keeps improving. No improvements were observed on either split after 400 steps.

GPT-4o and outperforms 1.5 Pro on standard math reasoning benchmarks (OpenAI, 2024). Overall, these results suggest that the reasoning flaws of cost-efficient LLMs may be obscured by high scores on prevalent math-reasoning benchmarks, underscoring the need to rethink current strategies for developing such models.

## 3.2 Instruction-Tuning Effects Vary Across LLM Sizes

We compare pretrained and instruction-tuned versions of small and large models in three LLMs families, namely Mistral, Llama-3 and Gemma2. Figure 5 illustrates this comparison, along with the performance gains from instruction-tuning, displayed above bars for each test set. On small models (top row), this comparison shows that current instruction-tuning is heavily optimized for GSM8K questions. Instruction-tuning leads to a significantly larger improvement on the original GSM8K test set than the compositional GSM test across model families. However, this trend does not apply or is reversed for larger LLMs (bottom row), despite using similar or identical data and training setup during instruction-tuning. Overall, these results suggest that smaller instruction-tuned LLMs exhibit systematic differences in their learning dynamics and generalization ability compared to their larger counterparts, complementing prior results for pretrained LLMs (Kaplan et al., 2020; Hernandez et al., 2021; Lotfi et al., 2024).

## 3.3 Math-Specialization Does Not Improve Reasoning Gap

Math-specialized LLMs are tailored to solve math reasoning problems. Such LLMs have an extensive data coverage for diverse mathematical domains, raising the question: do they generalize to held-out math reasoning tasks or overfit to standard benchmarks? To answer this question, we evaluated four state-of-the-art mathematical LLMs, namely NuminaMath-7B-CoT, Mathstral-7B, and Qwen2.5-Math-7B-IT and 72B-IT on GSM8K and compositional GSM (Figure 6).

We observe that these math-specialized LLMs exhibit reasoning gaps comparable to other models of similar size within our analysis. For instance, Qwen2.5-Math-7B-IT achieves above $80\%$ accuracy on difficult high-school competition level questions in MATH (Hendrycks et al., 2021), but solves less than 60% of the compositional grade-school math problems. This results is surprising, as most questions in the MATH test set are significantly more challenging than simply chaining two grade school questions together. Moreover, the large difference in compositional GSM between Qwen2.5-Math-IT 72B and 7B models, despite nearly similar GSM8K performance, reinforces our findings in Sec 3.1 that smaller LLMs exhibit systematic differences in their reasoning capabilities.

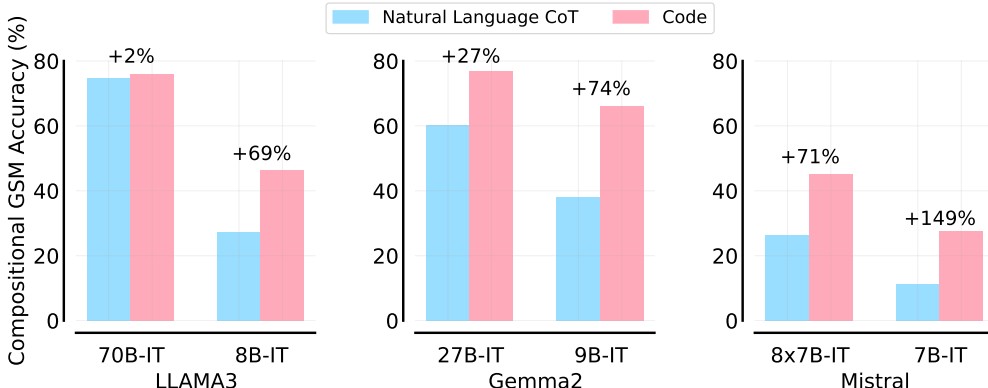

Figure 8: **Natural Language CoT** *versus* **Code:** Generating code to solve the problems helps in both settings of original test split and compositional GSM split. Numbers above bars represent relative improvements over natural language Chain-of-Thought (CoT) generation. Smaller models benefit more from generating code rather than natural language CoT to solve compositional GSM questions, further highlighting that smaller models demonstrate systematic differences in reasoning capabilities.

### 3.4 FINETUNING CAN LEAD TO TASK OVERFITTING

Supervised finetuning LLMs is a common strategy to improve their performance on reasoning tasks (Zelikman et al., 2022; Singh et al., 2023). In this section, we explore how it impacts the performance on compositional GSM. To do so, we finetune Gemma2 27B PT on the original GSM8K training dataset with human-written solutions, as well as synthetic data (Yuan et al., 2023), to identify any difference in the characteristics of these two sources. For synthetic data, we collect self-generated solutions that result in correct final answers for all GSM8K training queries. See Appendix C for details of data generation and training for this set of experiments.

When finetuning on either human or synthetic data (Figure 7), compositional GSM performance increases with some training (up to 100 steps), but drops with more training steps (400 steps) while GSM8K test performance keeps increasing, which suggests task-specific overfitting. Moreover, training on synthetic data generally leads to a higher performance on both GSM and compositional GSM. We did not observe further improvements on either test splits after 400 training steps. Based on this result, we hypothesize that the trend of using increasingly larger training datasets for overtraining small models beyond compute-optimal scaling (Sardana and Frankle, 2023; Touvron et al., 2023; Gadre et al., 2024) – often heavily composed of synthetic data (AI@Meta, 2024) – may primarily target performance on standard benchmarks, potentially at the expense of overall generalization and effectiveness across a wider range of tasks.

### 3.5 REASONING IN NATURAL LANGUAGE *versus* CODE

Breaking down natural language solutions into executable code can improve reasoning abilities of LLMs (Gao et al., 2023; Gou et al., 2023). To this end, we evaluate whether compositional problem-solving ability of LLMs improves when generating Python code instead of natural langauge CoT solutions. For code generation, we utilize a compositional 8-shot prompt (Appendix F), where the answers are written as two functions, one which solves the first question *solve_q1()*, and *solution()* which solves the second question with a $X = solve\_q1()$ line at the beginning.

We report our results in Figure 8 for three families of open-weight LLMs: LLAMA3-8B and 70B, Gemma2-9B and 27B, and Mistral 7B and Mixtral-$8 \times 7$B. We find that code generation generally improves performance on compositional GSM problems, albeit not uniformly. Comparing relative improvements, smaller models benefit substantially more from generating code solutions, again highlighting the systematic differences in their reasoning. While code generation may help reduce the gap for certain models, the primary aim of this study is not to "solve" compositional GSM as a benchmark. Further, often what matters most is not the final answer itself, but the interpretative process by which it was derived in natural language, making it applicable across a variety of contexts.

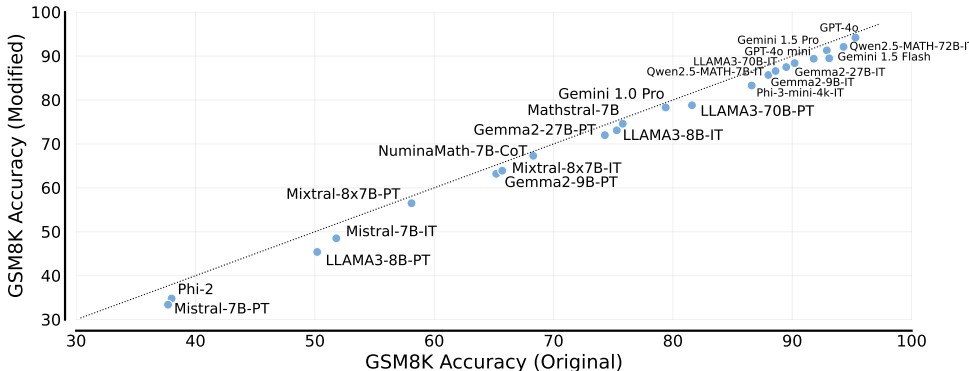

Figure 9: **Original (Q1) v.s. Modified GSM8K (Q2) test accuracy**. Most models are very close to the $x = y$ line, indicating that test set leakage is not a significant concern. Modified GSM8K questions are created by modifying a number in the original questions while ensuring that the new final answer remains a positive integer and is reasonably close to the original one.

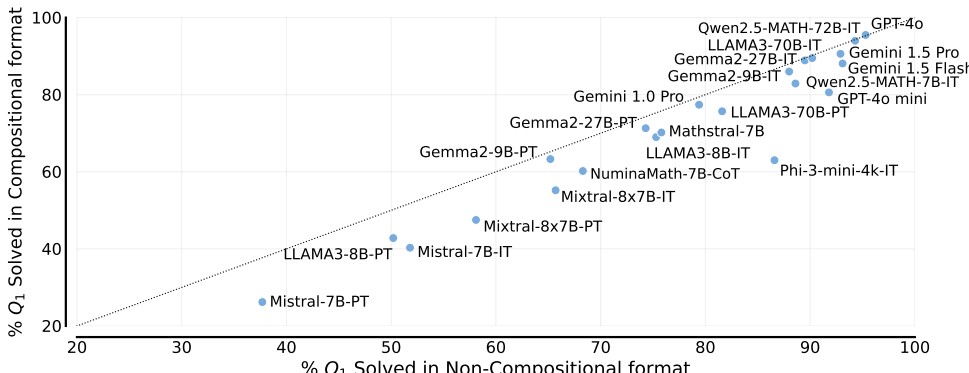

Figure 10: **Some LLMs get distracted easily:** Measuring models' ability to solve a question in the standard format (non-compositional) versus solving the same question as $Q_1$ in the compositional format. Models below the trend-line get distracted and cannot answer $Q_1$ in the compositional format even though solving it does not depend on solving any other question. The models generally adhere well to the output format provided in the 8-shot context, resulting in negligible instances of non-extractable answers.

### 3.6    WHY DO LLMS STRUGGLE WITH COMPOSITIONAL GSM?

**Does benchmark leakage cause degradation?**    Prior works hypothesize that test-data leakage (Xu et al., 2024; Golchin and Surdeanu, 2023) results in overestimating the mathematical capabilities of LLMs, as evidenced by performance degradation on GSM1K (Zhang et al., 2024), or functional variants of MATH problems (Srivastava et al., 2024). To this end, we evaluate how well LLMs perform on solving the modified GSM problems ($Q_2$ in compositional GSM) compared to original GSM8K test. Interestingly, Figure 9 shows that most LLMs obtain similar accuracy on modified GSM problems, suggesting that test-set leakage is not a major concern in our setup.

**Do LLMs Get Distracted Easily?**    Assuming an LLM answers a question correctly, it is expected that it would answer the same question correctly with additional context. However, Shi et al. (2023); Levy et al. (2024) find that LLMs can be easily distracted by irrelevant context. To this end, we study how often a model independently answers a question (from $Q_1$ set) correctly, and how often it answers the same question correctly in the compositional format, and report the results in Figure 10. Ideally, models should be on the $x = y$ line, but we observe that several models fall short of this expectation. Examining the responses from models with greater deviations from the trendline in Figure 10, we find that they often overlook important details, such as missing a reasoning step related to *each* in the question or omitting a arithmetic step when the question specifies *a month* or *per month*. This distraction is caused by the existence of a second question $Q_2$ in the prompt.

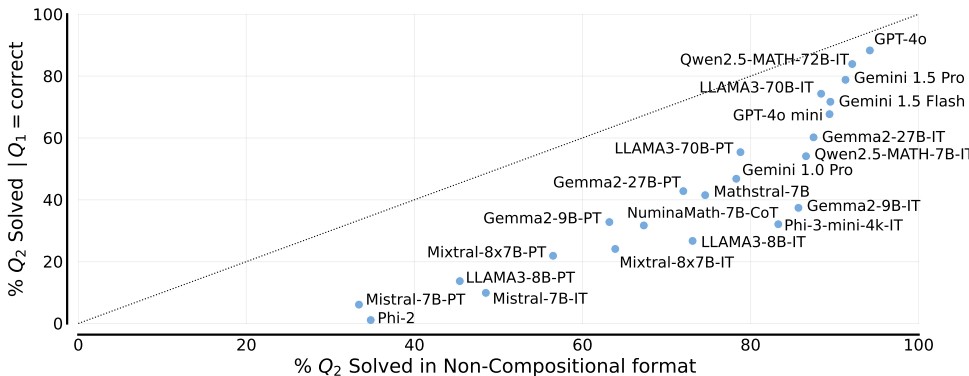

Figure 11: **Can models answer the second question if they have correctly answered the first one?** Here, we compare how often models are able to solve a question independently to how often they are able to solve them in the compositional format given that the first question is solved correctly. This is an alternate measurement of the compositional reasoning gap. If a model can solve a question independently, it should be able to solve it in a compositional setting given that the prerequisites are met. The gap from the diagonal line suggests that some models have overfit to the format of GSM8K type questions. While models may correctly answer the first question, they frequently makes subtle errors and miss key details when solving the second question.

Such failures lead to not being able to correctly answer $Q_1$, which subsequently impairs the models' ability to answer $Q_2$ correctly.

**Does Solving Question-1 Guarantee Solving Question-2?** Correctly solving Question-1 is a prerequisite to solve Question-2 in the compositional format. In Figure 11, we look at how often models are able to solve a question independently versus how often can they solve it given they have correctly solved the previous question in the compositional format. What remains for the model to do here is to substitute $X$ and solve $Q_2$. The deviation from the diagonal line indicates that certain models may have become too specialized in handling GSM8K-style questions, and are unable to answer a second question having generated the solution to the first question. Our qualitative analysis shows that when given two questions, the model might answer the first one correctly, but often makes subtle errors and overlooks details, leading to inaccurate reasoning and solution for the second question.

In Figure 12, we look at the capacity of models to solve two questions together in the context. We find that the distraction caused by $Q_1$ is limited when $Q_2$ is independent of it, but models have difficulty solving $Q_2$ accurately when it depends on the final answer of $Q_1$ even if $Q_1$ has been solved correctly. Overall, our results in Figure 11 and 12 align with the prior findings that when faced with multi-hop knowledge retrieval problems, LLMs can perform the first hop reasoning but not the second (Yang et al., 2024b; Press et al., 2023).

## 4 RELATED WORK

**Mathematical Reasoning Robustness.** Our work is heavily inspired by the study of robustness of math reasoning capabilities of LLMs via rewrites of GSM8K test queries (Zhang et al., 2024), or by employing functional variants of MATH problems (Srivastava et al., 2024). While these works argue for the possibility of test set leakage and memorization, our results in Figure 9 suggest that these issues are not a major concern in our setup. Others have investigated the robustness of math reasoning abilities of LLMs via adversarial examples (Anantheswaran et al., 2024; Li et al., 2024a), leakage estimation (Xu et al., 2024), semantic substitutions (Chen et al., 2023; Wang et al., 2023), and distractions within the context (Shi et al., 2023). In contrast to these works, our work focuses on two-hop grade school math reasoning, which we demonstrate does not always correlate with performance on math reasoning benchmarks. Please refer to Mondorf and Plank (2024); Ahn et al. (2024) for comprehensive surveys on LLM reasoning.

**Compositional Reasoning.** The ability of models to apply learned patterns to novel combinations of elements and generalize effectively has been studied extensively. Lake and Baroni (2018); Hupkes et al. (2020); Andreas (2020) have looked at seq2seq models' ability to compose known fractions

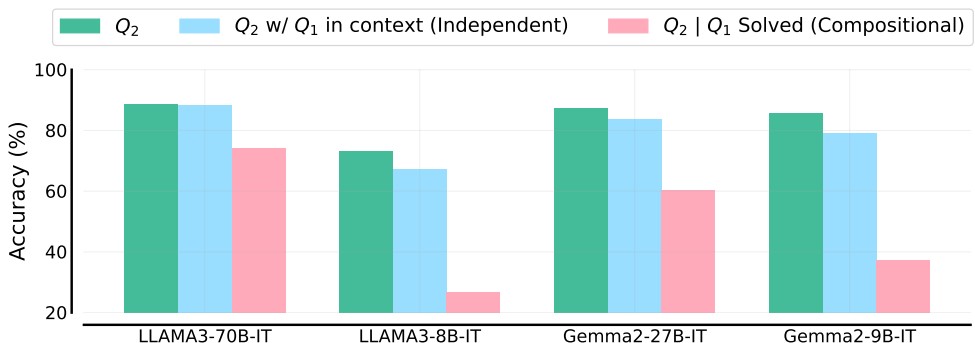

Figure 12: **Models Have the Capacity to Solve Two Questions Together:** Comparing models' ability to solve a question ($Q_2$) in three contexts: the standard format (non-compositional), with $Q_1$ in the context without depending on its answer, and in the compositional format given that $Q_1$ is solved correctly. The distraction from $Q_1$ in the context is minimal when $Q_2$ is independent of it. However, when $Q_2$ relies on the answer from $Q_1$, models struggle to solve $Q_2$ accurately, even if $Q_1$ has been answered correctly.

together into novel combinations in synthetic settings. More recently, the in-context compositional generalization of LLM reasoners has been examined (Hosseini et al., 2022; He et al., 2024; Yin et al., 2024; Kazemi et al., 2024). In contrast to such works, our work does not primarily emphasize compositional GSM as yet another benchmark; rather, it serves as a case study to highlight the differences in capabilities among various LLM reasoners. Press et al. (2023) find that the compositionality gap does not decrease as GPT-3 model size increases, which contrasts with our findings for frontier LLMs in Figure 4. Several studies have focused on adversarial attacks to evaluate multi-hop reasoning, emphasizing the prevention and examination of "shortcut learning" (Ding et al., 2021; Bhuiya et al., 2024; Ding et al., 2024). Instead, our work shows that LLMs can struggle with two-hop reasoning, even in non-adversarial scenarios. Others have focused on decomposing tasks into smaller skills for LLMs (Khot et al., 2023; Zhou et al., 2023); however, these approaches often necessitate prior knowledge of the specific skills or the use of specialized prompts for each task.

## 5  DISCUSSION

Our case study on compositional GSM demonstrates that most LLMs have still not "mastered" grade-school math reasoning, despite what their high performance on prevalent math reasoning benchmarks would suggest. Instead, LLMs may be exploiting superficial patterns in their training data, leading to an overestimation of their reasoning capabilities. Stress-testing LLMs with tasks like compositional GSM or counterfactual tasks is crucial for differentiating true understanding from superficial pattern matching (McCoy et al., 2023; Wu et al., 2023), highlighting the need for more "out-of-distribution" tasks to assess reasoning capabilities of LLMs (Shapira et al., 2023; Shah et al., 2024; Lewis and Mitchell, 2024; Li et al., 2024b).

A key finding of our work is that small and cost efficient LLMs, which are broadly accessible and crucial for real-world applications (Wan et al., 2024), exhibit larger reasoning gaps. Our analysis on these models uncovers their systematic differences in learning dynamics and flaws in reasoning capabilities, despite similar training settings and comparable performances on common benchmarks to larger, more expensive models. This raises the question of whether small and cost-efficient models are fundamentally limited in their ability to achieve such generalizations (Grosse et al., 2023).

Mathematical reasoning is inherently contextual and compositional, yet current evaluation methods fail to capture this complexity. Our compositional testing approach on grade-school math (GSM) reasoning has yielded significant insights, and we envision future work exploring the application of this testing approach to additional tasks and benchmarks, such as those from MATH (Hendrycks et al., 2021), or by extending our framework to multimodal reasoning problems. Our case study should not be viewed as an endpoint or merely as a tool for generating additional training data to "solve" compositional GSM problems, but as a catalyst to gain insights about the nature of reasoning of current LLMs as well as to re-evaluate how we assess "reasoning".

## 6 REPRODUCIBILITY STATEMENT

The prompts and preambles used for model generations are detailed in Appendix B, D, E, and F. Finetuning details related to Figure 7 can be found in Appendix C. Following Zhang et al. (2024), we do not plan to release our compositional GSM test publicly at this time to avoid potential issues of data contamination in the future.

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

# Appendices

## A    DISTRIBUTION OF FINAL ANSWER MAGNITUDES

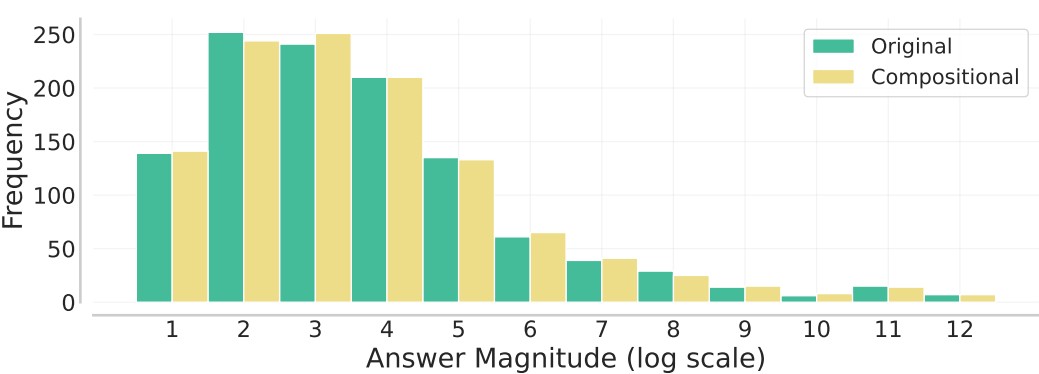

Figure A.1: **Distribution of final answer magnitudes** from the test set of original GSM8K and compositional GSM benchmark. The number modification in the compositional benchmark was done in a way to ensure that the new final answer is a positive integer not too far from the old answer. Our compositional GSM benchmark has a similar distribution of final answers.

## B    PROMPT PREAMBLES

---

**GSM8K Preamble**

```
I am going to give you a series of demonstrations of math Problems and
Solutions.  When you respond, respond only with the Solution of the
final Problem, thinking step by step.  At the end of the Solution, when
you give your final answer, write it in the form \The final answer is
ANSWER."
```

**Compositional GSM Preamble**

```
I am going to give you a series of demonstrations of compositional math
questions and solutions.  Respond by thinking step by step.  Solve the
first question and write the intermediate answer as \The Q1 answer is
ANSWER1.\ Then solve Q2.  At the end of the solution, when you give your
final answer, write it in the form \The final answer is ANSWER2."
```

---

## C    REJECTION FINETUNING DETAILS

Synthetic data was generated by prompting Gemma2 27B PT model with the 8-shot prompt in Appendix D to solve GSM8K training questions. We generated 10 solutions for each question in the original GSM8K training data, and only kept those solutions with a correct final answer. These model generation solutions were used to train the model. We evaluated intermediate checkpoints (at 50, 100 and 400 training steps) from both settings on GSM8K original test split and compositional GSM split.

## D  GSM8K 8-SHOT PROMPT

```
Q: There are 15 trees in the grove.  Grove workers will plant trees in
the grove today.  After they are done, there will be 21 trees.  How many
trees did the grove workers plant today?
A: There are 15 trees originally.  Then there were 21 trees after some
more were planted.  So there must have been 21 - 15 = 6.  The final
answer is 6.

Q: If there are 3 cars in the parking lot and 2 more cars arrive, how
many cars are in the parking lot?
A: There are originally 3 cars.  2 more cars arrive.  3 + 2 = 5.  The
final answer is 5.

Q: Leah had 32 chocolates and her sister had 42.  If they ate 35, how
many pieces do they have left in total?
A: Originally, Leah had 32 chocolates.  Her sister had 42.  So in total
they had 32 + 42 = 74.  After eating 35, they had 74 - 35 = 39.  The
final answer is 39.

Q: Jason had 20 lollipops.  He gave Denny some lollipops.  Now Jason
has 12 lollipops.  How many lollipops did Jason give to Denny?
A: Jason started with 20 lollipops.  Then he had 12 after giving some to
Denny.  So he gave Denny 20 - 12 = 8.  The final answer is 8.

Q: Shawn has five toys.  For Christmas, he got two toys each from his
mom and dad.  How many toys does he have now?
A: Shawn started with 5 toys.  If he got 2 toys each from his mom and
dad, then that is 4 more toys.  5 + 4 = 9.  The final answer is 9.

Q: There were nine computers in the server room.  Five more computers
were installed each day, from monday to thursday.  How many computers
are now in the server room?
A: There were originally 9 computers.  For each of 4 days, 5 more
computers were added.  So 5 * 4 = 20 computers were added.  9 + 20 is
29.  The final answer is 29.

Q: Michael had 58 golf balls.  On tuesday, he lost 23 golf balls.  On
wednesday, he lost 2 more.  How many golf balls did he have at the end
of wednesday?
A: Michael started with 58 golf balls.  After losing 23 on tuesday, he
had 58 - 23 = 35.  After losing 2 more, he had 35 - 2 = 33 golf balls.
The final answer is 33.

Q: Olivia has $23.  She bought five bagels for $3 each.  How much money
does she have left?
A: Olivia had 23 dollars.  5 bagels for 3 dollars each will be 5 * 3 =
15 dollars.  So she has 23 - 15 dollars left.  23 - 15 is 8.  The final
answer is 8.

Q: {question}
A:
```

# E COMPOSITIONAL 8-SHOT PROMPT

```
Let X be the answer to Q1:

Q1:  There are 15 trees in the grove.  Grove workers will plant trees
in the grove today.  After they are done, there will be 21 trees.  How
many trees did the grove workers plant today?

solve it and use the value of X to solve Q2.  Explain your answer step
by step.

Q2:  There are X students in Marissa's class.  Each student started
the year with 10 pencils.  After two months, 1/5 of the total pencils
in class were used.  At the end of the year, only 1/3 of the remaining
pencils were left.  How many pencils were left?

A: There are 15 trees originally.  Then there were 21 trees after some
more were planted.  So there must have been 21 - 15 = 6.  The Q1 answer
is 6.  Therefore X=6.  So there were 6 * 10 = 60 pencils in the class
at the start of the year.  After two months, 60 * 1/5 = 12 pencils were
used.  Thus, 60 - 12 = 48 pencils were left unused after two months.
Therefore, 48 * 1/3 = 16 pencils were left at the end of the year.  The
final answer is 16.

Let X be the answer to Q1:

Q1:  If there are 3 cars in the parking lot and 2 more cars arrive, how
many cars are in the parking lot?

solve it and use the value of X to solve Q2.  Explain your answer step
by step.

Q2:  Ingrid drinks X cups of water every day.  If there are 16 cups in a
gallon, how many gallons of water does she drink in 32 days?

A: There are originally 3 cars.  2 more cars arrive.  3 + 2 = 5.  The Q1
answer is 5.  Therefore X=5.  So Ingrid drinks 5 cups of water a day so
after 32 days she drinks 5 * 32 = 160 cups of water.  There are 16 cups
in 1 gallon so she drinks 160 / 16 = 10 gallons of water in 30 days.
The final answer is 10.
                  ⋮
Let X be the answer to Q1:

Q1:  {QUESTION_1}

solve it and use the value of X to solve Q2.  Explain your answer step
by step.

Q2:  {QUESTION_2}

A:
```

Some examples in the prompt are omitted due to space constraints. The remaining question-and-answer pairs follow the same format.

# F  COMPOSITIONAL CODE PROMPT

Some examples in the prompt are omitted due to space constraints. The remaining question and code solutions are written with the same format. The answers are provided through two functions: *solve_q1()*, which addresses the first question, and *solution()*, which answers the second question. The *solution()* function begins with a line *X = solve_q1()* to get the result from the first function.

```
Write two functions `solve_q1` and `solution` to solve Q1 and Q2
problems.

Let X be the answer to Q1:

Q1:  There are 15 trees in the grove.  Grove workers will plant trees
in the grove today.  After they are done, there will be 21 trees.  How
many trees did the grove workers plant today?

Q2:  There are X students in Marissa's class.  Each student started
the year with 10 pencils.  After two months, 1/5 of the total pencils
in class were used.  At the end of the year, only 1/3 of the remaining
pencils were left.  How many pencils were left?

A: The answer is
```
def solve_q1():
        """There are 15 trees in the grove.  Grove workers will plant
trees in the grove today.  After they are done, there will be 21 trees.
How many trees did the grove workers plant today?"""
        trees_initial = 15
        trees_after = 21
        trees_added = trees_after - trees_initial
        result = trees_added
        return result

def solution():
        """There are X students in Marissa's class.  Each student
started the year with 10 pencils.  After two months, 1/5 of the total
pencils in class were used.  At the end of the year, only 1/3 of the
remaining pencils were left.  How many pencils were left?"""
        X = solve_q1()
        num_students = X
        pencils_per_student = 10
        total_pencils = num_students * pencils_per_student
        pencils_left_after_two_months = total_pencils * (4/5)
        remaining_pencils = pencils_left_after_two_months * (1/3)
        result = remaining_pencils
        return result
```
            .
            .
            .

Let X be the answer to the following question:

Q1:  {QUESTION_1}

Q: {QUESTION_2}

A: The answer is
```

# G  ERROR ANALYSIS

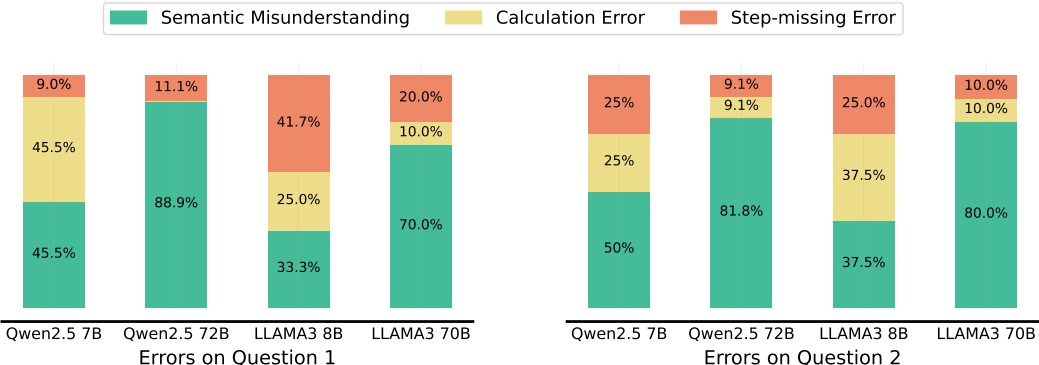

Figure G.1: **Error analysis on small and cost-efficient models**. We examined random subsets of question-responses from Qwen2.5-MATH-IT (7B and 72B) and LLAMA3-IT (8B and 70B) models and categorized mistakes into three groups of Semantic Misunderstanding, Calculation Error and Step-missing error following Zhong et al. (2024) either on $Q_1$ or $Q_2$. Smaller models tend to miss more reasoning steps, and make more calculation errors. Larger models make more semantic and logical errors, although the questions they fall short on are more difficult, and not necessarily the same questions. Total error rates on Compositional GSM for the models are 45.2, 15.8, 72.6 and 25.2 respectively.

### LLAMA3-8B-IT

**Q1**: Johnny is picking up the toys on the floor of his room. He'd dumped a lego boxed set with 500 pieces on the floor, and another one that had 3 times more pieces than the 500 piece one, and another one that had 1/4 the number of pieces. How many blocks does Johnny pick up if he picks up all the legos?

**Q2**: Last week the IRS received 5168 tax reports. On Monday and Tuesday they received a total of 1907 reports. On Thursday and Friday they received a total of X reports. How many reports did they receive on Wednesday?

**Response**: 500 pieces in the first box. 3 times more is 3 x 500 = 1500. 1/4 the number of pieces is 1/4 x 500 = 125. So in total there are 500 + 1500 + 125 = 2125 pieces. The Q1 answer is 2125. Therefore X=2125, so the IRS received 5168 - 1907 = 3261 reports on Wednesday. The final answer is 3261.

_________________________

Step-missing Error

### LLAMA3-70B-IT

**Q1**: Argo has 200 toys. He gives 40 toys to Alyssa, 80 to Bonnie, and 30 to Nicky. How many toys does Argo have now?

**Q2**: 90 single use contacts come in 1 box and will last Pete 45 days. Each box is $X and currently 10% off. If he buys 2 boxes of contact, how much will each pair of contacts cost?

**Response**: Argo started with 200 toys. He gave 40 to Alyssa, 80 to Bonnie, and 30 to Nicky. So he gave away 40 + 80 + 30 = 150 toys. 200 - 150 = 50. The Q1 answer is 50. Therefore X=50, so each box is 50 dollars. 10% off means 50 * .9 = 45 dollars. 2 boxes will cost 45 * 2 = 90 dollars. 90 dollars for 180 contacts. 90 / 180 = 0.5 dollars per contact. The final answer is 0.5.

_________________________

Semantic Misunderstanding Error

Figure G.2: **Example of errors for LLAMA3-IT 8B and 70B on compositional GSM.**

