# OpenReview forum: "Not All LLM Reasoners Are Created Equal"
_ICLR.cc/2025/Conference — Submitted to ICLR 2025_

### Official Review · Reviewer_iMpx · 2024-10-31

**Soundness:** 3
**Presentation:** 4
**Contribution:** 3
**Rating:** 6
**Confidence:** 4

**Summary:**

This paper focuses on compositional GSM, showing that most LLMs have not “mastered” grade-school math reasoning, despite high performance on common benchmarks. It introduces a two-hop test set where two linked questions must be solved simultaneously. Experiments reveal a significant performance drop compared to GSM8K.

**Strengths:**

The compositional evaluation demonstrates that high benchmark performance does not reflect flexible reasoning capabilities, which is vital for understanding LLM development.

The findings on fine-tuning leading to task overfitting are very interesting. It illustrates how fine-tuning on a single type of data, even within similar domains, can cause models to overfit, highlighting the importance of balancing downstream task performance with generalization.

**Weaknesses:**

The authors suggest that reasoning gaps arise from distractions of additional context and poor second-hop reasoning. Have you formally summarized the common types of mistakes? These errors might reflect issues with length generalization, coherence between questions, or reasoning patterns.


In the setup, did you combine different Q1s with a fixed Q2? It seems that the authors used a one-to-one mapping for constructing compositional test questions.

Regarding section 3.1, it appears that cost-efficient models show greater performance declines than high-cost models. Could this suggest that the general capabilities of smaller models hinder their understanding and solving of more complex questions? Have you analyzed the mistake patterns and behaviors of small versus large models?

In lines 369-377, whether the experiment is to compare between the accuracy of independently answering Q1 and the accuracy of determining the value of X in the compositional setting? How is the value of X parsed?

The experiments described in lines 369-415 are intriguing. Including qualitative case analyses on page 10 could provide a more detailed understanding of model mistakes.

**Questions:**

Please see the weakness section

---

> ### Author Response · Authors · 2024-11-18
>
> Thank you for the insightful and valuable comments.
>
> >  Have you formally summarized the common types of mistakes? Have you analyzed the mistake patterns and behaviors of small versus large models? Including qualitative case analyses on page 10 could provide a more detailed understanding of model mistakes.
>
> Thank you for the suggestion. We examined random subsets of question-responses from Qwen2.5-MATH-IT (7B and 72B) and LLAMA3-IT (8B and 70B) models and categorized mistakes into three groups of Semantic Misunderstanding, Calculation Error and Step-missing error (following Zhong et al. (2024)) either on Q1 or Q2. Here is the [figure](https://ibb.co/F66tL5r).
> We find that smaller models miss more reasoning steps, and make more calculation errors.
> Larger models make more semantic and logical errors, although the questions are more difficult. We also include an example of such errors in [this](https://ibb.co/61mNmcb) figure. Additionally, our analysis shows that less than 1% of the total responses have invalid or unextractable final answers across models.
>
> > In the setup, did you combine different Q1s with a fixed Q2? It seems that the authors used a one-to-one mapping for constructing compositional test questions.
>
> The selection of Q1 for a given Q2 is designed to ensure that the new final answer of Q2 remains a positive integer close to the original answer. We randomly choose from a pool of suitable Q1s that satisfy this condition.
>
> >  How is the value of X parsed?
>
> The compositional 8 shot prompt is shown in Appendix E. The value of X is parsed from the phrase “The Q1 answer is” in the outputs.
>
> ___
> References:
>
> Zhong et al. (2024), Achieving >97% on GSM8K: Deeply Understanding the Problems Makes LLMs Better Solvers for Math Word Problems

---

### Official Review · Reviewer_vgh2 · 2024-11-01

**Soundness:** 3
**Presentation:** 3
**Contribution:** 2
**Rating:** 5
**Confidence:** 4

**Summary:**

This explores the reasoning capabilities of large language models (LLMs) in solving grade-school math (GSM) problems, specifically focusing on compositional reasoning—where the solution to one problem depends on correctly solving another. The study introduces "Compositional GSM," a benchmark to test LLMs on pairs of related math problems, revealing significant reasoning gaps in most models. The study suggests that prevalent benchmarks might overestimate LLMs’ reasoning abilities and proposes compositional reasoning tasks as a more rigorous measure for evaluating true understanding in LLMs.

**Strengths:**

It also highlights multiple interesting findings, such as "small and cost efficient LLMs, which are broadly accessible and
crucial for real-world applications (Wan et al., 2024), exhibit larger reasoning gaps".

**Weaknesses:**

This paper tries to cover too many aspects without studying each in a detailed and comprehensive manner.

For example, it showed cost-efficient LLMs perform badly on compositional GSM8k, but does not try to probe into the reasons behind this with more detailed error analysis or case studies.

The study focuses on GSM8k vs two GSM8k questions chained together. Although GSM8k is a very important math reasoning benchmark, it fails to discuss other popular math reasoning datasets such as MATH which involve more complex and diverse reasoning. The authors have mentioned this as a future work, but having not discussed MATH weakens the overall contributions of the paper.

**Questions:**

Can you perform more detailed error analysis

---

> ### Author Response · Authors · 2024-11-18
>
> We appreciate your valuable feedback.
>
> > it showed cost-efficient LLMs perform badly on compositional GSM8k, but does not try to probe into the reasons behind this with more detailed error analysis or case studies. Can you perform more detailed error analysis?
>
> Our objective with this study is to use the compositional GSM task as a case study to highlight differences in reasoning capabilities of various LLMs, rather than to focus extensively on the task itself. We aimed to provide a broad perspective to inform future work and effort in this area.
>
> For your convenience, **we have included the results of our new analysis here (also in the revision):**
>
> **Error distribution on Question 1:**
>
> | Model                             | Semantic Misunderstanding  | Calculation Error               | Step-missing Error |
> | ---------------| ---------------: | ---------------: |---------------:|
> | LLAMA3 70B IT              |  70.0%                                   |  10.0%                              |  20.0%                   |
> | LLAMA3 8B IT                |  33.3%              	 	       |  25.0%                              | 41.7%                    |
> | Qwen2.5 MATH 72B IT   | 88.9%                                   | 0%                                     |11.1%                     |
> | Qwen2.5 MATH 7B IT     |  45.5%                                  | 45.5%                                | 9.0%                      |
>
>
>
>
> **Error distribution on Question 2:**
>
> | Model                              | Semantic Misunderstanding  | Calculation Error               | Step-missing Error |
> | ---------------| ---------------: | ---------------: |---------------:|
> | LLAMA3 70B IT              | 80.0%                                    |  10.0%                               |  10.0%                   |
> | LLAMA3 8B IT                |  37.5%              	 	       |  37.5%                              | 25.5%                    |
> | Qwen2.5 MATH 72B IT   | 81.8%                                   | 9.1%                                 | 9.1%                       |
> | Qwen2.5 MATH 7B IT     |  50%                                     |   25%                                | 25%                        |
>
>
> We analyzed random subsets of question-response pairs from LLAMA3-IT (8B and 70B) and Qwen2.5-MATH-IT (7B and 72B) models, categorizing errors into three types: Semantic Misunderstanding, Calculation Errors, and Step-Missing Errors, as outlined in Zhong et al. (2024), for both $Q_1$ and $Q_2$. Our findings indicate that smaller models tend to miss more reasoning steps and make more calculation errors. In contrast, larger models are more prone to semantic and logical errors, though the questions they struggle with are typically more complex and not necessarily the same across models.
>
> The overall error rates on Compositional GSM for the models are as follows:
>
> | Model                              | Error rate   |
> | ---------------| ---------------: |
> | LLAMA3 70B IT              |  25.2%        |
> | LLAMA3 8B IT                |  72.6%        |
> | Qwen2.5 MATH 72B IT   | 15.8%        |
> | Qwen2.5 MATH 7B IT     |  45.2%       |
>
> We also include an example of such errors in [this](https://ibb.co/61mNmcb) figure. Additionally, our analysis shows that less than 1% of the total responses have invalid or unextractable final answers across models.
>
>
> > it fails to discuss other popular math reasoning datasets such as MATH which involve more complex and diverse reasoning. The authors have mentioned this as a future work, but having not discussed MATH weakens the overall contributions of the paper.
>
> We chose GSM8K due to its widespread popularity, and the recent claims that LLMs have mastered grade-school math reasoning. We did highlight that our method is extensible to other tasks and benchmarks, including MATH, or even multimodal reasoning tasks. Our analysis on GSM8K has already revealed lots of new valuable insights into the capabilities and limitations of LLM reasoners. Notably, we find that smaller, cost-efficient, and math-specialized models show larger reasoning gaps (recognized by R-EFFT, R-vgh2), task specific finetuning can lead to task overfitting (recognized by R-EFFT and R-iMpx), and instruction-tuning affects models of different sizes differently (recognized by R-EFFT). If a model struggles with Compositional GSM problems, it's reasonable to expect it will also struggle with Compositional MATH questions given that MATH problems are more difficult.
>
> ___
> References:
>
> Zhong et al. (2024), Achieving >97% on GSM8K: Deeply Understanding the Problems Makes LLMs Better Solvers for Math Word Problems
>
> ___
>
> _We hope we've addressed your concerns and kindly ask that you reconsider your assessment. We're happy to continue the discussion if needed._

---

> > ### Author Response · Authors · 2024-11-28
> > **Gentle reminder for discussion**
> >
> > Dear reviewer vgh2,
> >
> > Since the discussion period draws to a close in the next few days, we were wondering if you have had a chance to go through our responses. Please let us know if your questions are addressed, we are happy to clarify anything remaining or any new questions. Thanks so much!

---

> > ### Author Response · Authors · 2024-12-01
> >
> > Dear reviewer vgh2,
> >
> > As the discussion period is approaching the closing date, we want to check in on whether our responses have addressed your concerns. We would be happy to discuss further.
> >
> > Authors

---

> > > ### Comment · Reviewer_vgh2 · 2024-12-02
> > > **Response to Rebuttal.**
> > >
> > > After thoroughly reviewing the rebuttal and the other reviewers’ comments, I still find that questions remain regarding what specific limitations in LLMs lead to their failure in compositional GSM in different settings and what could be the cause of such limitations despite the newly presented error analysis. Additionally, it is not entirely clear if the proposed benchmark indeed measures compositional generalization. Therefore, I decide to keep my score.

---

> > > > ### Author Response · Authors · 2024-12-02
> > > >
> > > > Thank you for your valuable feedback.
> > > >
> > > > We ran the error analysis as per reviewer 3RUa's and your request, categorizing model errors following Zhong et al., (2024). We were pleased to hear that it addressed reviewer 3RUa’s concerns. Additionally, we conducted new experiments with reverse order of questions in compositional GSM (as [suggested](https://openreview.net/forum?id=T8PzwgYgmn&noteId=fyJWPAjUJD) by Reviewer iMpx) which strengthened our findings.
> > > >
> > > > Our analysis indicates that models struggle with such complex contexts. The cause of these errors, across the diverse and large set of LLMs we tested, presents an interesting avenue for future work. The key factor affecting performance appears to be model size/capacity/cost, as even math-specialized models we tested, such as Qwen, MetaMath and MAmmoTH2, with extensive math related data and compute, faced challenges.
> > > >
> > > > > Additionally, it is not entirely clear if the proposed benchmark indeed measures compositional generalization
> > > >
> > > > We did not make this claim in the paper. Rather, we view compositional GSM as a litmus test or case study that highlights the systematic differences in reasoning abilities among LLMs, despite their similar training setups and comparable scores on standard benchmarks. Tasks like our compositional GSM, along with concurrent work such as Mirazeh et al. (2024) on GSM-Symbolic (which complements our results), are essential for stress-testing LLMs and revealing flaws that may not become apparent from traditional benchmarks.
> > > >
> > > > We hope this addresses your remaining concern.
> > > >
> > > >
> > > >
> > > > ----
> > > > References:
> > > >
> > > > Zhong et al. (2024), Achieving >97% on GSM8K: Deeply Understanding the Problems Makes LLMs Better Solvers for Math Word Problems
> > > >
> > > > Mirzadeh et al. (2024), GSM-Symbolic: Understanding the Limitations of Mathematical Reasoning in Large Language Models

---

> ### Author Response · Authors · 2024-11-27
> **Gentle reminder**
>
> Thank you for your time and valuable insights; we hope we've addressed your concerns and comments, and just wanted to send a gentle reminder.

---

### Official Review · Reviewer_EFFT · 2024-11-03

**Soundness:** 3
**Presentation:** 3
**Contribution:** 3
**Rating:** 5
**Confidence:** 5

**Summary:**

The paper studies how well Large Language Models (LLMs) can handle compositional grade-school math problems by evaluating their performance on pairs of GSM8K questions where the answer to the first question is needed to solve the second. The main finding is that while many LLMs perform well on individual GSM8K problems, they show significant "reasoning gaps" when solving these compositional pairs. The paper positions itself not just as introducing another benchmark, but as a case study revealing systematic differences in LLM reasoning capabilities that aren't apparent from standard benchmark performance. The work suggests that current evaluations may overestimate LLMs' true mathematical reasoning abilities, particularly for smaller and more cost-efficient models that are important for practical applications.

**Strengths:**

1. Introduction of Compositional GSM, a new evaluation approach that chains two GSM8K test questions together, requiring models to correctly solve both in sequence

2. Comprehensive evaluation of various LLMs, including Gemini, Gemma2, LLAMA3, GPT, Phi, Qwen2.5, and Mistral families

3. Several important empirical findings:
- Most models show a clear performance gap between standard GSM8K and compositional problems
- Smaller, cost-efficient, and math-specialized models show larger reasoning gaps
- Instruction tuning affects models of different sizes differently
- GSM8K-specific finetuning can lead to task overfitting
- The reasoning gaps stem from distraction by additional context and poor second-hop reasoning, rather than test set leakage

4. Analysis showing that code generation helps improve performance on compositional problems, particularly for smaller models

**Weaknesses:**

- My biggest concern is the gap between the motivation of the study and the proposed evaluation method. The authors propose the Compositional GSM as a tool to evaluate LLM reasoning, but don't sufficiently validate that the task actually measures real compositional reasoning ability. While they show poor performance on the second question even when the first is solved correctly, they don't establish whether this definitively indicates a reasoning gap versus other potential issues like prompt sensitivity, instruction following ability, or other format issues.
- Further clarification on the motivation of the study. Even though we see the dropping performance, what does it mean in reality? In what kind of real scenarios, the users would have such evaluation settings like that? How does the performance transform into the real challenge we would like to solve?
- Lack of concrete solutions for the problem. Though the authors discuss several potential ways that lead to the phenomenon, they do not try to propose solutions to solve the issue. For example, it is still unclear whether the downgraded performance is caused by the sensitivity or the format of the new test setting. Would it be possible to quickly try some simple fine-tuning? For example, could you repurpose the GSM8K training set to something like how you constructed the test set? Would such a simple fine-tuning method solve the issue?
- The evaluation was only conducted on GSM8K dataset, which limited the conclusion's validity to other reasoning scenarios.
- The paper claims to examine math-specialized models but tests a relatively narrow set (Numina-7B, Mathstral-7B, Qwen2.5). Some more open-source and open-weight math models might be added for a more comprehensive analysis. e.g., deepseek-math, MetaMath, MAmmoTH, etc.

**Questions:**

See Weaknesses

---

> ### Author Response · Authors · 2024-11-18
>
> Thank you for your insightful feedback.
>
> > While they show poor performance on the second question even when the first is solved correctly, they don't establish whether this definitively indicates a reasoning gap versus other potential issues like prompt sensitivity, instruction following ability, or other format issues.
>
> Figure 12 compares models’ abilities to answer a question in different contexts. In the setting where Q1 and Q2 are both in the context, but are independent, models show minimal issues. This is an almost identical format to the setting where Q2 depends on Q1. Therefore models do not encounter any formatting issues in these settings. Additionally, our analysis shows that less than 1% of the total responses have invalid or unextractable final answers across models. As noted in Section 3, some models required a preamble before the 8-shot prompt to properly follow instructions. We tested both configurations (with and without the preamble) and reported the best performance for each model.
>
> > Even though we see the dropping performance, what does it mean in reality? In what kind of real scenarios, the users would have such evaluation settings like that?
>
> Many real-world use cases involve compositing multiple skills, and if LLMs struggle with even the simplest of these scenarios, they are likely to encounter significant challenges with more complex ones. This evaluation also complements testing on more challenging benchmarks, which, based on our results with math-specialized models, may have contamination issues.
>
> > Could you repurpose the GSM8K training set to something like how you constructed the test set? Would such a simple fine-tuning method solve the issue?
>
> Finetuning on specific examples, like our compositional GSM, might help models address this particular issue, but that would miss the border purpose of our work. One could design additional similar tests to uncover other shortcomings. Relying on specific finetuning to cover every possible case is not a feasible approach.
>
> > The evaluation was only conducted on GSM8K dataset, which limited the conclusion's validity to other reasoning scenarios.
>
> While this is a reasonable limitation, our work nonetheless reveals important and surprising differences between models that perform similarly on GSM8K.  Notably, influential works such as Zhang et al. (2024) and, concurrently, Mirazeh et al. (2024), among many, examine the grade-school mathematical reasoning capabilities of LLMs--an active area of research. If a model struggles with Compositional GSM problems, it is reasonable to expect it will also face difficulties with similar compositional setup of more difficult benchmarks such as MATH.
>
> > The paper claims to examine math-specialized models but tests a relatively narrow set (Numina-7B, Mathstral-7B, Qwen2.5). Some more open-source and open-weight math models might be added for a more comprehensive analysis. e.g., deepseek-math, MetaMath, MAmmoTH, etc.
>
> We selected Qwen2.5 Math (as well as Numina & Mathstral) models because they represent the state-of-the-art math specialized performance for their sizes. We’ve extended our analysis to MetaMath-Mistral-7B and MAmmoTH2-8B-Plus both underperforming Qwen2.5 in both settings.
> Here are the results:
>
> | Model | GSM8K Acc | Compositional GSM Acc |
> | -------- | -------------  | -------------------------- |
> | MetaMath-Mistral-7B | 74.6 | 11.1                             |
> | MAmmoTH2-8B-Plus	|	  79.7		|            33.8 |
>
> ___
>
> References:
>
> Zhang et al. (2024), A Careful Examination of Large Language Model Performance on Grade School Arithmetic
>
> Mirazeh et al. (2024), GSM-Symbolic: Understanding the Limitations of Mathematical Reasoning in Large Language Models
> ___
>
> _We hope that we've addressed most of your concerns. If so, we kindly ask that you reconsider your assessment. We would be glad to continue the discussion if needed._

---

> > ### Comment · Reviewer_EFFT · 2024-11-23
> > **Follow up**
> >
> > Thank you for your detailed response to my review. While I appreciate the effort you’ve made in addressing my comments, I still find that my main concern regarding the **motivation of the study** remains unresolved.
> >
> > You have clarified some aspects of the study, such as the experimental setup and the use of additional models. However, I would like to restate and expand on my initial concern:
> >
> > ### **Motivation Gap:**
> > The motivation behind evaluating compositional reasoning via the proposed Compositional GSM is still unclear in terms of practical relevance and its connection to real-world scenarios. Specifically:
> >
> > - While you argue that "many real-world use cases involve compositing multiple skills," the specific relevance of the Compositional GSM task to such scenarios remains underexplored. Could you provide concrete examples where users face settings similar to your evaluation? For instance, what practical workflows or applications hinge on an LLM's ability to solve sequential math problems in this manner?
> >
> > - How does the performance drop observed in Compositional GSM map to a broader class of challenges in LLM reasoning? Without tying the observed phenomenon to a clear, tangible use case or broader reasoning problem, the motivation feels disconnected from practical challenges that users or researchers face.
> >
> > - The study highlights reasoning gaps, but how critical are these gaps in real-world applications? For instance, do these gaps represent a fundamental limitation of LLMs, or could they be mitigated by better prompting or other practical techniques? Addressing this point would significantly strengthen the motivation.
> >
> > I will keep my initial ratings.

---

> > > ### Author Response · Authors · 2024-11-23
> > >
> > > Thank you for your reply.
> > >
> > > > Could you provide concrete examples where users face settings similar to your evaluation?
> > >
> > > In many real-word scenarios, solving complex problems requires breaking them down into smaller, more manageable components. Often, these components take the form of sequential (math) problems, where each step builds on the results of the previous one. Examples of use cases and scenarios include financial planning, tax calculation, data processing and analysis (such as SQL queries), engineering designs and structural integrity assessments, and even code generation–where one function relies on the output of another one. In all these cases, the ability to solve problems in a **step-by-step, interdependent manner** is essential for achieving accurate and effective results.
> > >
> > > Given our analysis, if many frontier LLMs struggle with even the simplest compositional math problems, how can we rely on them to handle more complex scenarios like those described above?
> > >
> > > > could they be mitigated by better prompting or other practical techniques?
> > >
> > > While specific prompting techniques may help address this particular issue, relying on them would again miss the broader goal of our work. As previously mentioned, one could design similar tests to uncover further shortcomings, but continuously adapting prompts to cover every possible case is neither scalable nor a sustainable solution. The goal should be to improve the model's inherent reasoning abilities, not to depend on workaround strategies for each new challenge.
> > >
> > > LLMs must be capable of managing this logical flow and accurately composing math or reasoning tasks. Ultimately, our analysis seeks to test the robustness of LLMs while also identifying and addressing gaps in their capabilities.

---

> > > > ### Author Response · Authors · 2024-11-29
> > > >
> > > > Thank you again for your valuable feedback. We are pleased to hear that we have addressed 4 out of 5 weaknesses in your initial review, and that you find our experiments comprehensive and our findings important.
> > > >
> > > >
> > > > To further address your follow up, the primary motivation of this paper is to conduct a case study exploring whether frontier LLMs reason differently. As highlighted in Section 5, the introduced compositional GSM task should not be viewed merely as an endpoint.
> > > >
> > > > Through this case study, we demonstrate repeatedly that LLMs exhibit systematic differences in their reasoning capabilities. Surprisingly, we find that smaller, cost-efficient LLMs, which are critical for deployment, may be overestimated in their reasoning abilities. This is despite having similar training settings and scores on standard benchmarks. Tasks like compositional GSM or concurrent work by Mirazeh et al. (2024) on GSM-Symbolic (to which our results are complementary) are crucial for stress-testing LLMs and uncovering flaws that may not be evident in traditional benchmarks.
> > > >
> > > > Even in our latest results (mentioned in reply to Reviewer iMpx) on reverse order of questions, we again observed these systematic differences in LLAMA 3 and Gemma 2 families.
> > > >
> > > >
> > > > This kind of compositional math is an easy version of composing skills in real world settings. For instance, in a medical scenario, an LLM would need to analyze a patient's symptoms, medical history, lab results, etc to arrive at a comprehensive diagnosis. In such a scenario, even the smallest misstep in reasoning, such as overlooking a crucial symptom, or misinterpreting statistical correlations can lead to significant issues.
> > > >
> > > > We hope this addresses your follow up. Thank you.

---

> > > > > ### Author Response · Authors · 2024-12-02
> > > > >
> > > > > Thank you once again for feedback. We’re pleased to share that both reviewers iMpx and 3RUa have acknowledged that their concerns have been addressed, resulting in a boost to their assessments.
> > > > >
> > > > > We hope we’ve also resolved your remaining concern regarding the primary motivation of our paper. Furthermore, in response to Reviewer iMpx’s suggestion, our [new experiments](https://openreview.net/forum?id=T8PzwgYgmn&noteId=fyJWPAjUJD) on the reverse order of questions further support our title "Not All LLM Reasoners Are Created Equal".
> > > > >
> > > > > | Model                    | Compositional | Compositional reversed | Independent | Independent reversed      |
> > > > > |------------------------|-------------:|--------------------------------:|--------------:|--------------------------------:|
> > > > > | LLAMA3 70B IT    | 74.8                |  75.8                                 | 88.4               | 86.2                                |
> > > > > | LLAMA3 8B IT      | 27.4                |  26.3                                 | 67.4               | 65.5                                |
> > > > > | Gemma2 27B IT   | 60.3                | 61.2                                  | 83.8               | 85.5                                |
> > > > > | Gemma2 9B IT     | 38.0                | 37.3                                  | 79.1               | 76.8                                |
> > > > >
> > > > > To summarize, reversing the order of questions in the compositional setting (dependent) increases performance for large models, but slightly decreases performance for smaller models, further supporting that smaller, cost-efficient models reason differently.
> > > > >
> > > > > We are happy to discuss this further if you'd like.

---

> > > > > > ### Author Response · Authors · 2024-12-03
> > > > > >
> > > > > > Dear reviewer EFFT,
> > > > > >
> > > > > > The discussion period is now over. Thank you for the interactions.
> > > > > >
> > > > > > If we've addressed your final concern, please consider increasing your score.

---

> ### Author Response · Authors · 2024-11-27
> **Gentle reminder**
>
> Thank you for your time and valuable insights. We hope to have addressed your questions regarding real-world scenarios, and we would be glad to continue the discussion if needed.

---

### Official Review · Reviewer_3RUa · 2024-11-04

**Soundness:** 4
**Presentation:** 4
**Contribution:** 2
**Rating:** 6
**Confidence:** 4

**Summary:**

This paper introduces a two-hop version of GSM8K, called compositional GSM8K, where two questions from the original test set are combined into one after making minimal changes. The analysis reveals significant reasoning gaps in LLM performance, particularly in smaller, cost-efficient, and math-specialized models. Instruction tuning improves performance on compositional GSM, but to a lesser extent than on the standard split, highlighting a greater challenge.

**Strengths:**

1. The proposed compositional GSM8K is a straightforward and effective method for testing the two-hop math reasoning abilities of LLMs.

2. The paper is easy to follow. The experiments are comprehensive, covering various types of models. The analysis explores various potential factors that could lead to weaker performance, such as model sizes, instruction tuning, fine-tuning, and using code as a format. It sufficiently supports their central claim regarding the deficiencies of LLMs in compositional mathematical reasoning.

**Weaknesses:**

1. As noted by the authors in the related work section, there are already several benchmarks for assessing the robustness of LLMs in math reasoning. Although this paper includes extensive experiments, the conclusion that LLMs struggle with multi-hop reasoning is not particularly surprising to me.

2. The two-hop QA format appears to be a minimal approach for testing mathematical compositional reasoning abilities. I was hoping for a more intricately designed benchmark, similar to SCAN or CFQ.

**Questions:**

The authors argue in Line 477 that "This raises the question of whether small and cost-efficient models are fundamentally limited in their ability to achieve such generalizations" and suggest we should rethink how we assess “reasoning". Can you elaborate on your insights and how we can achieve this?

---

> ### Author Response · Authors · 2024-11-18
>
> Thank you for your thoughtful and insightful comments.
>
> > there are already several benchmarks for assessing the robustness of LLMs in math reasoning. Although this paper includes extensive experiments, the conclusion that LLMs struggle with multi-hop reasoning is not particularly surprising to me.
>
> Our analysis on compositional GSM goes beyond simply highlighting LLMs’ challenges with multi-hop reasoning. We uncover notable performance differences among models that show near-identical results on the original GSM8K benchmark. This variation in performance is particularly intriguing, and in many cases, surprising. For example, we find that smaller, more cost-efficient LLMs exhibit distinct differences in their learning dynamics and reasoning shortcomings, despite having nearly identical training setups and comparable results on standard benchmarks. Rather than introducing yet another benchmark, our goal is to provide insights into the nuanced behaviors of different models, fostering deeper exploration and inspiring future research.
>
> > The two-hop QA format appears to be a minimal approach for testing mathematical compositional reasoning abilities. I was hoping for a more intricately designed benchmark, similar to SCAN or CFQ.
>
> We selected GSM8K for this study due to its widespread recognition as well as the recent claims suggesting that LLMs have successfully mastered grade-school math problem-solving. Our compositional GSM study offers a more realistic and streamlined approach, and has already revealed lots of new valuable insights. GSM8K questions already require a good amount of intricate reasoning from the model, so a composition of these questions will inherently involve similar complexity and reasoning abilities by design.
>
>
> > The authors argue in Line 477 that "This raises the question of whether small and cost-efficient models are fundamentally limited in their ability to achieve such generalizations" and suggest we should rethink how we assess “reasoning". Can you elaborate on your insights and how we can achieve this?
>
> We noted that mathematical reasoning is inherently contextual and compositional, but existing evaluation methods tend to focus on task performance, which can be a superficial measure of a model’s reasoning abilities. For instance, GPT-4o and GPT-4o mini have very similar GSM8K accuracies (95% vs 92%), but very different compositional GSM performances (88% vs 68%).
> Our work highlights these limitations and contributes to the deeper exploration of LLMs’ reasoning capabilities. To truly assess reasoning, we must go beyond simple performance scores and consider factors such as the model's ability to handle complexity, context and compositionality. This shift in focus could provide more meaningful insights into the depth and nuance of LLMs’ reasoning capabilities.
>
> ___
>
> _We hope that most of your concerns have been addressed and, if so, we would appreciate it if you could reconsider your assessment. We’d be happy to engage in further discussions._

---

> > ### Comment · Reviewer_3RUa · 2024-11-25
> > **reply to authors**
> >
> > Thank you for your response.
> >
> > >For example, we find that smaller, more cost-efficient LLMs exhibit distinct differences in their learning dynamics and reasoning shortcomings, despite having nearly identical training setups and comparable results on standard benchmarks.
> >
> > From Figure 5, I can see that LLAMA3-8B has greater gains on Comp GSM by instruction tuning, although the absolute score is lower than Gemma2-9B. What leads to such different learning dynamics?
> >
> > I also found [1] studied the robustness of evaluating GSM8K and reached a similar conclusion that smaller models suffer from greater performance drops (Table 4). If smaller models are unrobust, it is natural to assume that they would be worse at two-hop reasoning because of the lack of stability of reasoning processes. If this paper aims to provide a deeper understanding, I think the authors can perform some error annotation and analysis on where models make mistakes.
> >
> > [1] [GSM-Plus: A Comprehensive Benchmark for Evaluating the Robustness of LLMs as Mathematical Problem Solvers](https://aclanthology.org/2024.acl-long.163) (Li et al., ACL 2024)

---

> ### Author Response · Authors · 2024-11-24
> **Gentle Rebuttal Reminder**
>
> Thank you once again for your valuable feedback on our work. We've carefully worked to address your comments and questions. Are there any further questions or concerns we should discuss?

---

> ### Author Response · Authors · 2024-11-25
>
> Thank you for your reply.
>
> The Gemma family is trained using a different approach compared to the LLaMA 3 family, with notable differences not only in training recipes but also in architecture. Key differences in their training recipes include variations in post-training procedures, datasets, reward models for RLHF, and choices between distillation or training from scratch, among other factors. For more detailed information on each family's training methodology, please refer to their respective technical reports (Gemma team et al., 2024; AI@Meta, 2024). These distinct training choices contribute to the differences in performance observed across existing benchmarks, including our compositional GSM task.
>
> > I also found [1] studied the robustness of evaluating GSM8K
>
> To further analyze the differences of small and cost-efficient models, we examined random subsets of question-responses from Qwen2.5-MATH-IT (7B and 72B) and LLAMA3-IT (8B and 70B) models and categorized mistakes into three groups of Semantic Misunderstanding, Calculation Error and Step-missing error (following Zhong et al. (2024)) either on Q1 or Q2. Here is the [figure](https://ibb.co/F66tL5r).
>
> We find that smaller models tend to miss more reasoning steps, and make more calculation errors. Larger models make more semantic and logical errors, although the questions they fall short on are more difficult, and not necessarily the same questions. We also include an example of such errors in [this](https://ibb.co/61mNmcb) figure. We will include these figures in the paper as well.
>
> ----
>
> References:
>
>
> (AI@Meta, 2024) Llama 3 model card. 2024. URL https://github.com/meta-llama/
> llama3/blob/main/MODEL_CARD.md
>
> (Gemma team et al., 2024) T. M. Gemma Team, C. Hardin, R. Dadashi, S. Bhupatiraju, L. Sifre, M. Riviere, M. S. Kale, `
> J. Love, P. Tafti, L. Hussenot, and et al. Gemma. 2024. doi: 10.34740/KAGGLE/M/3301.

---

> > ### Author Response · Authors · 2024-11-28
> > **Follow up reply**
> >
> > Thank you again for your valuable feedback. We are glad that all your concerns in the initial review have been resolved.
> >
> > For your convenience, **we have included the results of the new analysis regarding your follow up question here:**
> >
> > **Error distribution on Question 1:**
> >
> > | Model                             | Semantic Misunderstanding  | Calculation Error               | Step-missing Error |
> > | ---------------| ---------------: | ---------------: |---------------:|
> > | LLAMA3 70B IT              |  70.0%                                   |  10.0%                              |  20.0%                   |
> > | LLAMA3 8B IT                |  33.3%              	 	       |  25.0%                              | 41.7%                    |
> > | Qwen2.5 MATH 72B IT   | 88.9%                                   | 0%                                     |11.1%                     |
> > | Qwen2.5 MATH 7B IT     |  45.5%                                  | 45.5%                                | 9.0%                      |
> >
> >
> >
> >
> > **Error distribution on Question 2:**
> >
> > | Model                              | Semantic Misunderstanding  | Calculation Error               | Step-missing Error |
> > | ---------------| ---------------: | ---------------: |---------------:|
> > | LLAMA3 70B IT              | 80.0%                                    |  10.0%                               |  10.0%                   |
> > | LLAMA3 8B IT                |  37.5%              	 	       |  37.5%                              | 25.5%                    |
> > | Qwen2.5 MATH 72B IT   | 81.8%                                   | 9.1%                                 | 9.1%                       |
> > | Qwen2.5 MATH 7B IT     |  50%                                     |   25%                                | 25%                        |
> >
> >
> > We analyzed random subsets of question-response pairs from LLAMA3-IT (8B and 70B) and Qwen2.5-MATH-IT (7B and 72B) models, categorizing errors into three types: Semantic Misunderstanding, Calculation Errors, and Step-Missing Errors, as outlined in Zhong et al. (2024), for both $Q_1$ and $Q_2$. Our findings indicate that smaller models tend to miss more reasoning steps and make more calculation errors. In contrast, larger models are more prone to semantic and logical errors, though the questions they struggle with are typically more complex and not necessarily the same across models.
> >
> > The overall error rates on Compositional GSM for the models are as follows:
> >
> > | Model                              | Error rate   |
> > | ---------------| ---------------: |
> > | LLAMA3 70B IT              |  25.2%        |
> > | LLAMA3 8B IT                |  72.6%        |
> > | Qwen2.5 MATH 72B IT   | 15.8%        |
> > | Qwen2.5 MATH 7B IT     |  45.2%       |
> >
> > Below is an example for a missed step by LLAMA3 8B IT, shown in `[  ]`:
> > ```
> > Q1: Johnny is picking up the toys on the floor of his room.  He'd dumped a lego boxed set with 500 pieces on the floor, and another one that had 3 times more pieces than the 500 piece one, and another one that had 1/4 the number of pieces.  How many blocks does Johnny pick up if he picks up all the legos?
> >
> > Q2: Last week the IRS received 5168 tax reports. On Monday and Tuesday they received a total of 1907 reports. On Thursday and Friday they received a total of X reports. How many reports did they receive on Wednesday?
> >
> > Response: 500 pieces in the first box. 3 times more is 3 x 500 = 1500. 1/4 the number of pieces is 1/4 x 500 = 125. So in total there are 500 + 1500 + 125 = 2125 pieces. The Q1 answer is 2125.
> > Therefore X=2125, so the IRS received 5168 - 1907 = 3261 reports on Wednesday. [    ]. The final answer is 3261.
> > ```
> >
> > Here is an example of semantic misunderstanding error from LLAMA3 70B IT, shown in `[  ]`:
> > ```
> > Q1: Argo has 200 toys. He gives 40 toys to Alyssa, 80 to Bonnie, and 30 to Nicky. How many toys does Argo have now?
> >
> > Q2: 90 single use contacts come in 1 box and will last Pete 45 days.  Each box is $X and currently 10% off.  If he buys 2 boxes of contact, how much will each pair of contacts cost?
> >
> > Response: Argo started with 200 toys. He gave 40 to Alyssa, 80 to Bonnie, and 30 to Nicky. So he gave away 40 + 80 + 30 = 150 toys. 200 - 150 = 50. The Q1 answer is 50.
> > Therefore X=50, so each box is 50 dollars. 10% off means 50 * .9 = 45 dollars. 2 boxes will cost 45 * 2 = 90 dollars. 90 dollars for 180 contacts. [90 / 180 = 0.5 dollars per contact]. The final answer is 0.5.
> > ```
> >
> > —-----
> >
> > Since the discussion period draws to a close please let us know if your follow up questions are addressed as well. We are happy to clarify any new questions. Thank you so much.

---

> > > ### Author Response · Authors · 2024-12-01
> > >
> > > Dear reviewer 3RUa,
> > >
> > > As the discussion period nears its closing date, we wanted to check in whether our responses have addressed your concerns. We’re happy to continue the conversation if needed.
> > >
> > > Authors

---

> > > ### Comment · Reviewer_3RUa · 2024-12-02
> > > **response to authors**
> > >
> > > Thanks for your follow-up. The newly added experiments addressed my concerns, so I have increased my overall rating to 6.

---

> ### Author Response · Authors · 2024-12-02
>
> We are happy to hear that all your concerns have been addressed, and we sincerely appreciate you increasing your overall rating.
>
> -- Authors

---

### Meta-Review · Area_Chair_ikkS · 2024-12-21

**Metareview:**

This paper studies LLM reasoning capabilities through a novel compositional GSM8K benchmark that chains two math problems together. While the empirical findings reveal interesting performance gaps between models, especially for smaller cost-efficient LLMs, the reviewers raised valid concerns about insufficient analysis of error patterns, limited scope focusing only on GSM8K, and unclear practical relevance. The paper's strengths lie in comprehensive model evaluations and discovery of systematic differences in reasoning abilities. However, the lack of deeper error analysis, connection to real-world applications, and extension beyond GSM8K limit its contribution. Decision: Marginally reject due to these limitations outweighing the interesting empirical findings.

**Additional Comments On Reviewer Discussion:**

The discussion centered on three main concerns: 1) Insufficient error analysis - authors provided new categorization of mistakes but reviewers found it inadequate for compositional settings; 2) Limited practical motivation - authors cited examples like financial planning but failed to fully convince reviewers of real-world relevance; 3) Narrow focus on GSM8K - authors argued their findings would likely extend to harder tasks but did not demonstrate this empirically. While authors made good faith efforts to address concerns through additional experiments and analysis, the core limitations around practical relevance and depth of analysis remained inadequately resolved according to multiple reviewers (EFFT, vgh2, iMpx), leading to the rejection recommendation.

---

### Decision · Program_Chairs · 2025-01-22

Reject